adolescents; depression; developing countries; health risk behaviors; qualitative research; preventive health services

**Corresponding author:**
Brandon A. Kohrt;
Email: bkohrt@gwu.edu

*Kieling and Fisher are joint senior authors.

# No prediction without prevention: A global qualitative study of attitudes toward using a prediction tool for risk of developing depression during adolescence

Brandon A. Kohrt[1] , Syed Shabab Wahid[2] , Katherine Ottman[1], Abigail Burgess[3], Anna Viduani[4], Thais Martini[4], Silvia Benetti[4], Olufisayo Momodu[5], Jyoti Bohara[6], Vibha Neupane[6], Kamal Gautam[1,6] , Abiodun Adewuya[7] , Valeria Mondelli[8,9] , Christian Kieling[4,10]* and Helen L. Fisher[3,11]*

[1]Center for Global Mental Health Equity, Department of Psychiatry and Behavioral Health, George Washington University, Washington, DC, USA; [2]Department of Global Health, Georgetown University, Washington, DC, USA; [3]Social, Genetic & Developmental Psychiatry Centre, Institute of Psychiatry, Psychology and Neuroscience, King's College London, London, UK; [4]Department of Psychiatry, Universidade Federal do Rio Grande do Sul, Porto Alegre, RS, Brazil; [5]Department of Psychiatry, Lagos Island General Hospital, Lagos, Nigeria; [6]Transcultural Psychosocial Organization Nepal (TPO Nepal), Baluwatar, Kathmandu, Nepal; [7]Department of Behavioural Medicine, Lagos State University College of Medicine, Lagos, Nigeria; [8]Department of Psychological Medicine, Institute of Psychiatry, Psychology & Neuroscience, King's College London, London, UK; [9]National Institute for Health and Care Research Maudsley Biomedical Research Centre, South London and Maudsley NHS Foundation Trust and King's College London, London, UK; [10]Child & Adolescent Psychiatry Division, Hospital de Clínicas de Porto Alegre, Porto Alegre, RS, Brazil and [11]ESRC Centre for Society and Mental Health, King's College London, London, UK

## Abstract

Given the rate of advancement in predictive psychiatry, there is a threat that it outpaces public and professional willingness for use in clinical care and public health. Prediction tools in psychiatry estimate the risk of future development of mental health conditions. Prediction tools used with young populations have the potential to reduce the worldwide burden of depression. However, little is known globally about adolescents' and other stakeholders' attitudes toward use of depression prediction tools. To address this, key informant interviews and focus group discussions were conducted in Brazil, Nepal, Nigeria and the United Kingdom with 23 adolescents, 45 parents, 47 teachers, 48 health-care practitioners and 78 other stakeholders (total sample = 241) to assess attitudes toward using a depression prediction risk calculator based on the Identifying Depression Early in Adolescence Risk Score. Three attributes were identified for an acceptable depression prediction tool: it should be understandable, confidential and actionable. Understandability includes depression literacy and differentiating between having a condition versus risk of a condition. Confidentiality concerns are disclosing risk and impeding educational and occupational opportunities. Prediction results must also be actionable through prevention services for high-risk adolescents. Six recommendations are provided to guide research on attitudes and preparedness for implementing prediction tools.

## Impact Statement

Prediction tools can be used to determine who is at high risk of developing a health condition in the future. Despite rapidly developing prediction models, there is a lack of knowledge about whether the public wants to use these tools to learn about their risks for future mental health conditions. This study explores the attitudes of adolescents, parents, teachers, health-care practitioners and policymakers, across diverse cultural settings in Brazil, Nepal, Nigeria and the United Kingdom, toward the use of a predictive tool for identifying the risk of future depression during adolescence. By highlighting the varying attitudes toward predictive psychiatry in different cultural contexts, the study underscores the necessity of culturally sensitive approaches to implement prediction tools in clinical care and public health. The study emphasizes the importance of making prediction models not only accurate but also actionable. Any risk prediction should safeguard adolescents' privacy, promote mental health literacy and ensure that preventive services are available whenever telling young people that they have a high risk of developing a mental health condition. The research outlines a pathway for developing new prediction tools in collaboration with adolescents and other stakeholders. The recommendations from this study can guide development of policies that facilitate the responsible use of prediction models. This is especially important given the growing potential of prediction through use of

artificial intelligence used alongside health records, online personal information and data collected from personal mobile devices. By incorporating perspectives from low- and middle-income countries as well as high-income countries, this study ensures that the benefits of predictive psychiatry are not limited to high-income nations but are extended to regions where the majority of the world's adolescents reside.

## Introduction

Risk detection to predict future development of a health condition is often considered a silver bullet in public health because it enables prevention and early intervention to reduce morbidity and mortality (Jonas et al., 2021, Livingston et al., 2020, Loomans-Kropp and Umar, 2019, Tapia-Conyer et al., 2017). Risk prediction refers to the process of estimating the probability that a specific individual will develop a certain condition over a defined period (Fuller and Flores, 2015). This information typically is used in clinical settings to inform selection and timing of interventions. For mental health conditions, there has long been an interest in prediction, with growing clinical and public health application for early intervention in psychosis (Fusar-Poli et al., 2017, Salazar de Pablo et al., 2021). The potential for prediction models in psychiatry is rapidly expanding to other conditions and populations (Bernardini et al., 2017, Tan et al., 2023). This includes prediction of depression among adolescents. In the United Kingdom (UK), a risk prediction calculator was developed and validated in two populations for adolescents with a parental history of depression (Stephens et al., 2023).

Despite growing potential for clinical and public health application of predictive models, there is less information on public attitudes toward the desire to know personal risk (Lawrie et al., 2019, Siddaway et al., 2020). Prior studies on attitudes toward predictive psychiatry have focused mostly on high-income countries among individuals with a family history of mental health conditions or populations already experiencing subsyndromal symptoms (Alder et al., 2013, Bui et al., 2014, Stephens et al., 2023, Welsh and Tiffin, 2011). However, the experience of risk prediction among individuals with a family history has differences from the expanding opportunities for risk prediction in the general public. For example, individuals with a family history are likely to have more awareness of the condition and already recognize the possibility of developing the condition themselves (Austin, 2020). With risk prediction for the general public, the individual may not have familiarity with the lived experience of the condition, and they may not consider themselves at risk (Walter and Emery, 2005, Wolff et al., 2010). The PredictD initiative with adults in Spain is one of the few initiatives that has collected qualitative feedback on attitudes from primary care patients, inclusive of persons who did not have a family history of depression, and general physicians regarding willingness to have depression risk assessed (Bellón et al., 2014, Moreno-Peral et al., 2019).

In low- and middle-income countries (LMICs), it is unknown whether the public, health-care providers and others are willing to participate in risk assessment for future mental health conditions. Access to mental health services is extremely limited (World Health Organization, 2022). Only one out of 27 people with depression in lower middle-income countries has minimally adequate depression treatment compared to 1 out of 4 in high-income countries (Thornicroft et al., 2017). Because few people receive care, knowledge of family history of depression is rarely available or accurate. Therefore, risk prediction would rely on willingness of the general public and health-care providers to participate in risk assessment. To address the gap in prediction tools that would be feasible in LMICs and the lack of information around attitudes toward such tools, the Identifying Depression Early in Adolescence (IDEA)

research consortium developed a risk prediction calculator for adolescent depression (Kieling et al., 2019, Kieling et al., 2021, Piccin et al., 2024). The IDEA risk score calculator (IDEA-RS) is based on easily obtained social and demographic variables, and it does not rely on an adolescent's knowledge of family history of depression (Rocha et al., 2021). In contrast to the high costs needed for genetic testing, neuroimaging and big data needed for other prediction models, the IDEA-RS can be feasibly implemented in health-care settings, schools and communities in LMICs and other low-resource settings (Wahid et al., 2021). Development of the IDEA-RS followed the stages of developing predictive tools in medicine (McGinn et al., 2000): Stage 1: selecting variables; Stage 2: validation in a single site; Stage 3: validation in different sites; and Stage 4: establishing the impact on clinical practice, with the latter stage focusing on acceptability of use.

For Stages 1 and 2, in a longitudinal cohort in Brazil, a constellation of 11 risk factors collected directly from adolescents at age 15 years achieved a 0.71 discriminative ability (area under the curve, AUC) to predict depression onset at 18 years of age: a 3-year prediction period (Rocha et al., 2021). For Stage 3, multiple replication analyses were conducted in high-income countries: UK (AUC = 0.59), United States (USA; AUC = 0.63) and New Zealand (AUC = 0.63); middle-income countries: in South America with a second population in Brazil (AUC = 0.69) and in Sub-Saharan Africa with an urban sample in Nigeria (AUC = 0.62); and one low-income country: a South Asian population affected by a humanitarian emergency: former child soldiers in Nepal (AUC = 0.73) (Brathwaite et al., 2020, Brathwaite et al., 2021, Caye et al., 2022, Cunha et al., 2023, Rocha et al., 2021). Evaluating performance of the IDEA-RS across settings is important because of the pitfalls of nontransportability of risk prediction findings across contexts due to differences in the measurement of depression and risk factors, cultural and experiential differences in the meaning of depression and risk factors, context and structure of health-care systems and different case mixes in study samples (Kohrt and Patel, 2020, Moons et al., 2012, Steel et al., 2014, Viduani et al., 2024).

In preparation for Stage 4, establishing the impact on clinical practice, we wanted to understand the attitudes of adolescents, health-care professionals and other stakeholders toward sharing personal information in a prediction calculator and obtaining risk of future depression. Do adolescents want to know if they are at risk of depression? How would parents, teachers, social workers, health-care professionals and policymakers interpret and use this information? What are the implications for educational and health systems to provide services for adolescents at higher risk? In order to consider using the IDEA-RS with the general public, this included needing to understand attitudes of adolescents and stakeholders where there is a lack of information about family history of depression. To answer these questions, we conducted a qualitative study in settings where the accuracy of the tool had been evaluated: three LMICs (Brazil, Nigeria and Nepal) and one high-income country (the UK). As with the quantitative testing of the IDEA-RS across settings, it is important to address the diversity of perspectives across contexts. The qualitative findings are intended to inform implementation of depression risk prediction to

minimize adverse impacts and optimize benefits for young people, families, schools and health systems around the world.

## Methods

### Overview

Full details of the IDEA qualitative study design are available in a published protocol (Wahid et al., 2020). In accordance with the Consolidated Criteria for Reporting Qualitative Research guidelines (Tong et al., 2007), all methodological details are provided in Supplemental File 1.

### Study settings

This qualitative study was implemented in urban settings of four countries where analyses on the predictive validity of the IDEA-RS had been evaluated using longitudinal data sets (Brathwaite et al., 2020, Brathwaite et al., 2021, Caye et al., 2022, Rocha et al., 2021). Brazil is an upper middle-income country in which 22% of the population are adolescents. Mental health care in Brazil is integrated into the Unified Health System, including specialized psychosocial centers for child and adolescent services, which are characterized as understaffed and under-resourced to address mental health needs (Marchionatti et al., 2023, Pimentel et al., 2023). The Brazil study site was the southern city of Porto Alegre, population of 1.3 million. The second site, Nepal, was a low-income country at the time of the study, and it lacks a nationwide child mental health-care program (Rai et al., 2021). Adolescents make up 24% of Nepal's population (National Statistics Office, 2021). The qualitative study in Nepal was conducted in the urban Kathmandu valley, population of 1.6 million. The third site, Nigeria, is a lower middle-income country and the most populous country in Africa with over 41 million adolescents, comprising 23% of the total population (National Bureau of Statistics, 2022). Adolescent mental health services are limited to urban regions. The qualitative study was conducted in the urban setting of Lagos, population of 16 million. The final site, the UK, is a high-income country in which adolescents comprise 18% of the population (Association of Young People's Health, 2024). Child and Adolescent Mental Health Services are provided by the National Health Service (2018). Qualitative interviews were conducted in the urban setting of London, population of 8.9 million.

### Study design and tools

This multisite qualitative study utilized key-informant interviews (KIIs) and focus-group discussions (FGDs) with adolescents and other relevant stakeholders to explore views on the feasibility and acceptability of an online risk calculator for adolescent depression. An initial deductive, cross-country interview guide was created and piloted with different stakeholders across each site. Interviewers maintained debriefing forms that captured observations of the context, emerging researcher insights, and observations on the structural implementation of the interviews. Data from preliminary KIIs and debriefing forms were used to revise the guides for contextual sensitivity and modify content for type of respondent (e.g. adolescents vs. policymakers). All subsequent interviews were constructed around a mock-up of the risk calculator.

### Description of risk calculator mock-up

The mock-up of the IDEA-RS online calculator was done in Google Forms and included 11 domains of questions reflecting the factors included in the risk prediction model validated in each site (see Supplemental File 2). The domains in the validated risk calculator include (1) female sex, (2) member of a minority group in that setting, (3) school underachievement, (4) relationship with friends, (5) relationship with mother, (6) relationship with father, (7) relationship between parents, (8) alcohol and drug use, (9) getting into fights, (10) running away from home and (11) history of child maltreatment. An English version was used for the UK and Nigeria, Brazilian Portuguese for Brazil and Nepali for Nepal.

The presentation of the calculator was modeled after widely used online risk calculators for diabetes (American Diabetes Association, 2024). In the mock calculator, the questions about risk factors were followed by a mock result presenting high-risk and low-risk outcomes with generic information to seek prevention programs if high-risk or engaging in healthy behaviors if low-risk. Interview participants were able to click through the calculator entering dummy information. Participants were shown both the high-risk and low-risk results screens regardless of what information they entered. No score was calculated for the participants as this was intended to walk them through the process but not provide any person-specific result that could have adverse effects. FGDs included demonstrating the risk calculator to stakeholder groups.

### Sample, recruitment and data collection

Participants were recruited using purposive sampling for key stakeholders (Patton, 2014). This included adolescents, some of whom had lived experience of depression, and parents or other caregivers. Professionals responsible for adolescent health, education and welfare were recruited including clinicians, researchers, teachers and social workers. Government policymaker recruitment included national and regional decision-makers in ministries of health and regional health coordinating bodies. These are individuals responsible for budgetary decisions, drafting legislation and government health workforce composition and allocation. Sample size rationale is provided in the full protocol (Wahid et al., 2020). Adolescents were not recruited in Nigeria or the UK because of concerns about using the risk calculator directly with young people.

KIIs and FGDs lasted approximately 60–90 min. KIIs and FGDs were audio recorded with consent, plus assent when interviewing minors. Audio recordings from sites were translated into English for data analysis with the exception of Brazil, where qualitative analysis was conducted in Portuguese. Code summaries were then created in English, with selected quotes translated by the Brazilian research team.

### Data analysis

Data analysis was conducted using the Framework Approach (Gale et al., 2013, Smith and Firth, 2011). A codebook was developed based on deductive themes. The codebook was revised to include site-specific findings and inductive codes. Transcripts were coded in NVivo version 12 (QSR International, 2017) by nine IDEA researchers from Brazil, Nepal, UK and USA. A minimum inter-rater reliability of 0.7 (Cohen's *kappa*) was required by coders before moving on to independent coding (McHugh, 2012). Summaries of each code were written to encapsulate themes with supporting quotes and researcher insights. The current qualitative

analysis focuses on attitudes toward use of a prediction calculator based on the IDEA-RS. Analyses focusing on other qualitative domains have been published previously (Burgess et al., 2022, Ottman et al., 2022, Viduani et al., 2021, Viduani et al., 2022, Wahid et al., 2022). Code query outputs relevant to the analyses described here are available in Supplemental File 3.

## Results

### Demographics

The study was conducted from December 2018 through January 2020. There were 241 participants (23 adolescents, 45 parents, 47 teachers, 48 health-care workers and 78 other stakeholders) across the four sites (65 = Brazil, 71 = Nepal, 45 = Nigeria and 60 = UK), see Table 1 for the main qualitative findings. For each qualitative finding below, the countries (BR-Brazil, NG-Nigeria, NP-Nepal, UK) and number of respondents are provided in parentheses. Additional information on frequencies for themes and subthemes by country is provided in Supplemental File 4.

### Benefits of using a risk calculator

The qualitative interviews revealed potential benefits for using a risk calculator to predict future onset of adolescent depression (Box 1). Benefits focused on the opportunity to engage in self-care to prevent depression (BR, NP, $n = 9$). Brazilian and Nepali adolescents also stated that being high risk could encourage help-seeking (BR, NP, n = 17). British social workers and other providers indicated the risk calculator could improve their ability to objectively identify adolescents needing services and support (UK, n = 21). They explained that this would improve upon current deployment of services, which tends to be subjective. Policymakers and social workers in Nigeria and Nepal noted that risk information can help to plan for the population's future (NG, NP, n = 58). Nigerian social workers and teachers highlighted that identifying at-risk adolescents, then supporting them, would lead to reduction of suffering at a societal level (NG, n = 4). However, potential benefits were dependent upon three attributes of the risk calculator implementation: first, the results need to be understandable for lay persons; second, adolescent data need to be kept confidential; and third, actionable support needs to be provided for at-risk adolescents (Figure 1). Each of these three attributes are described below.

### Understandable

Understandable referred to ensuring the prediction tool was delivered in the context of educating adolescents and parents about mental health, such as through depression literacy programs. Respondents reported growing awareness about mental health and depression among youth compared to older generations (NG, UK, $n = 24$). However, they emphasized the need to raise awareness about the treatability of depression (BR, NP, UK, $n = 34$). Participants recommended using the risk calculator only when accompanying it with depression education information, such as through a website, app, pamphlets and community awareness programs (NP, NG, UK, $n = 28$). A respondent in Nigeria encouraged educating adolescents about the negative impact of depression on families and society in order to motivate adolescents to pursue prevention activities (NG, $n = 1$).

Participants raised a concern that both adolescents and adult stakeholders would interpret the purpose of the risk calculator as detection of current depression, such as a screening tool for depression (BR, NG, NP, UK, $n = 31$). This misunderstanding was confirmed by the observation that many respondents' initial answers during the qualitative interviews suggested that they thought the tool would tell them their current depression status. To address this, some respondents recommended simple illustrations and language to quantify the likelihood of risk and the accuracy of the prediction (BR, NG, NP, UK, $n = 35$), and they suggested including language to explain the meaning of a low-risk result. Respondents were concerned that adolescents receiving low-risk results would falsely assume they were guaranteed to never develop depression. Respondents in Nepal and the UK suggested that risk numbers need to be calibrated to the local context for adolescents to have accurate information to interpret their likelihood of developing depression (NP, UK, $n = 19$).

The tool was recommended to be available in multiple languages in Nigeria and Nepal (NG, NP, $n = 7$). They also said it needs to be simple enough for adolescents with low literacy. Across countries, respondents suggested that health-care practitioners or teachers should be trained to explain the risk calculator, explain depression, clarify having a condition versus risk of a condition and help interpret results (BR, NP, NG, UK, $n = 66$). These respondents also recommended that trained counselors should administer the tool and deliver the results because they could make the information understandable in a way that minimized distress for youth identified as high risk.

**Table 1.** Qualitative study participants by country.

| Type of participant | Brazil: Porto Alegre | Nepal: Kathmandu | Nigeria: Lagos | United Kingdom: London | Total |
|---|---|---|---|---|---|
| Adolescents | 11 | 12 | - | - | 23 |
| Parents and caregivers | 12 | 18 | 3 | 12 | 45 |
| Health-care practitioners* | 12 | 11 | 13 | 12 | 48 |
| Social workers | 12 | 14 | 12 | 12 | 50 |
| Teachers and school staff** | 12 | 10 | 13 | 12 | 47 |
| Policymakers | 6 | 6 | 4 | 12 | 28 |
| Total per site | 65 | 71 | 45 | 60 | 241 |

*Includes psychiatrists, psychologists and child and adolescent mental health specialists.
**Includes teachers, school counselors and school nurses.

**Box 1.** Example quotes for key themes on attitudes toward use of a risk calculator in Brazil, Nepal, Nigeria and the United Kingdom.

### 1. Benefits of risk calculator

#### 1.1. Improved self-care

"People that have low risk will feel good and maybe are going to try helping others. And, people with high risk maybe will take care of themselves better." (Depressed adolescent, Brazil, BR-4)

"It is a faster, more practical, individual way. It is a personal answer for people to identify and become aware of themselves and seek help, which in general does not happen on its own. I think it will give the person an idea because it will identify them: ah, I have these options, I can get help …" (School principal, Brazil, BR-12)

**"**You'll get to know if you are in [high risk] or not. [Then] get help on what we should do, and get more help on preventing it … Like, what can we do at this stage, whom to share the information with, and how can we be protected?" (Depressed adolescent, Nepal, NP-39)

"If I find out I have high risk of depression, I will be more careful. I will try to self-motivate. I will try to avoid tension. I will try to build a positive attitude … I will share with others … If others don't have suggestions for me on how to prevent depression, I will search for information on the internet. No matter what, we need to have a positive attitude." (Non-depressed adolescent, Nepal, NP-35)

"Eh, [we need a] sane society, a progressive society. Because the people we are talking about now mainly are the future of tomorrow. The sanity of this nation tomorrow depends on them. If you start planning for them now at this stage, you're preparing them for a better future." (Social worker, Nigeria, NG-02)

#### 1.2. Objectively identifying adolescents in need of services

"I always appreciated when I did get information about students' home lives, and I knew what was going on. That made my life a lot easier … So, in that sense, it would be really useful to have something that, you know, was evidenced-based, and you didn't have to do your own kind of qualitative interpretation of the data." (Social worker, UK, UK-39)

### 2. Understandable

#### 2.1. Need for mental health literacy about depression

"[Using the risk calculator] depends upon the choice of words. When [a result is provided as] high risk, content like information that depression can be cured should be included, rather than just on normal things like nutrition and help-seeking. Otherwise, if there isn't information about treating depression, the fear will be there." (Depressed adolescent, Nepal, NP-43)

"We all know that there are very few parents in Nepal who sufficiently understand these things. That is why we need to inform the parents, as well the students, about what [depression] is." (School nurse, Nepal, NP-26)

"[We need] something that kind of gets parents on board, erm, in a way that doesn't get a lot of knee jerk resistance to it? Kind of like vaccinations have developed … And in a way that just makes it a part of looking after your health, just a sort of part of your preventative care, you know? Get your five a day, get your mental health screening, you know?" (Parent, UK, UK-82)

#### 2.2. Distinguishing between having a condition and risk of a condition

"The ethical implications are major. We have to consider that it's really hard for humans to work with probability, we are not good 'probabilists' [sic] in general, and when we … we say that someone is at high risk, we must always remember that, first, we could be very wrong, and second, we need to be careful about self-fulfilling prophecies, right? Oh, so you are at high risk of becoming depressed? Well, then you might indeed become depressed. So, we have to evaluate if this doesn't happen, what's the effect? What if it generates a collateral effect? These things are very difficult to evaluate, and we will need clinical studies." (Psychiatrist, Brazil, BR-9)

"We need to speak positively with them and tell them that there is a chance that they will not suffer from it … This does not mean that they will have this disease for sure, but they are only at risk. We should tell them about the things that they need to avoid in order to reduce their chances of getting it. If we are able to counsel them as such, then they will be able to relate it a little." (School nurse, Nepal, NP-26)

"Well, same as any screening tool. You know, you get false positives, you have unnecessary morbidity. You have the pathologization of somebody who doesn't need to be pathologized, particularly at the age and state that they're at. It's better to have a narrative which is just about life circumstance than there's something wrong with you." (Health-care worker, UK, UK-21)

### 3. Confidential

#### 3.1. Confidentiality

"Because [the risk calculator] has facts that happen to most adolescents, and they probably don't tell anyone. But if it has confidentiality, if you tell them that you are not going to tell anyone, the person is going to trust you and probably will answer everything with the truth. They won't lie anywhere, won't deny anything, because they will be trusting this." (Depressed adolescent, Brazil, BR-40)

"… In communities like ours, people generally have negative attitudes toward mental health problems or mental illness. Therefore, I don't think this needs to be revealed to others. Like, if teachers know about this then he could say this person is like this and that. If parents and guardians come to know then they will also start to say this and that. This also can happen in the community. Therefore, this should be done from individual level … I think we should maintain the privacy." (Health-care worker, Nepal, NP-04)

"It should not be a self-administered questionnaire that means the adolescent cannot administer by themselves. So, it should be health worker, or teacher, counsellor or anybody that has been trained to identify, should be administered by those people." (Teacher, Nigeria, NG-16)

#### 3.2. Sharing results with stakeholders

"I see more of us caregivers using it than teenagers. … You will access it, you will have a sense, right, of what will be the result. But, then [the adolescent] will not have much contact with it. I even think that it would not be good for teenagers to access it, too, it would be something more for adults because … sometimes he will end up believing that it is something that is not, and create paranoia, something." (Social worker, Brazil, BR-5)

"According to their results, they need to sit down with their guardians and talk to them. It means that their results have to be disclosed in a minor form first. And then we should disclose the major implications when their guardian is with them. The adolescents will also feel that they need to tell their guardians now. They will feel that it is a decision that they have taken themselves." (Health-care worker, Nepal, NP-27)

"It will be good [to share with parents]. If we share with [an adolescent], she will have tension. If shared with guardians, they will try making their child feel good. Their child will feel that people care about them. A kind of positive attitude is developed. If shared with others [and not the adolescent], she won't know that she is likely to have depression and then she will not develop tension. Others will show her love knowing this and try to care. Due to this, the negative feelings clear out." (Non-depressed adolescent, Nepal, NP-35)

"It'll be better for the child not to know if he or she is at high risk or low risk. Let it be known to the person attending to the child. So, the person attending to the child will know how to – and because sometimes if they know, it might trigger some other things, you understand. So, that's why it is even better sometime for them not to – fine they may be filling the form, but when it comes to that aspect where it's going to calculate if they are at high risk or low risk, it should be best known the person that is attending to them. Because normally children are very sensitive, it is what they see, they react to, it is what they hear, they react on. So, I believe keeping it away from them or the society will really help a lot." (Social Worker, Nigeria, NG-06)

### 4. Actionable

#### 4.1. Prevention services need to be available

"What happens is the following. So, I have my – my diagnosis of high risk for depression. But then it is a lot about the luck of this adolescent, that he will look for another resource … And then [the adolescent] goes to somewhere and then goes to another one, they end up on a waitlist. It could even be dangerous, because the person is already in a high risk of depression." (Teacher, Brazil, BR-10)

"At the same time, it is a matter of practical reality that at the moment, at least in Brazil, but I believe that in the world, we do not have – there is a great lack of mental health professionals, let alone in our country, so we barely and barely manage to identify cases already in depression, already with the syndrome, and make the appropriate treatment. There is a lack of medication, a lack of psychotherapy, in droves, and a lack of case identification, a lack of looking at the cases we already have. So, it is very difficult for us to achieve. Of course, prevention is still important because it possibly reduces this overload. But at the same time, in terms of Brazil, it is very difficult for us to imagine yet." (Parent, Brazil, BR-9)

"We have to provide them the service if needed, as this is in our ethics as well." (Parent of depressed adolescent, Nepal, NP-01)

"Let's say the risk is identified. The person may be in high risk, medium risk, or low risk. After knowing that, where will the person go and find the service? We talked so much about how there is no service available for depression. That's what we say in public health: if you don't have the solution, you should not see the problem." (Policymaker, Nepal, NP-052)

"… at first, when you showed it to me, I thought, oh, like, yeah, it's not very complicated to answer. But then, what do you do, what is the next step, like, when you have the, ok you are in high risk, what do you do with that information." (Parent, UK, UK-97)

"I believe that it would only be ethical really to roll it out to people if it then would trigger some support. Because, I think it's really, erm, dangerous to get into a situation where you've got a young person saying this happened to me and effectively, kind of, not being listened to and actioned." (School worker, UK, UK-68)

### Confidential

Regarding consent for using the calculator, opinions varied with some participants suggesting that adolescents should independently consent (NG, NP, UK, *n* = 34). Others in Nigeria and Nepal suggested adolescents provide assent and guardians provide consent (NG, NP, *n* = 11). A few participants in Nigeria and the UK said only the guardian is responsible for consent (NG, UK, *n* = 10). A policymaker in Nigeria recommended that the calculator be administered during annual school-based health screenings, for which parents provide consent (NG, *n* = 1). Respondents suggested informing parents about this being done, but not everyone endorsed requiring parental consent (NP, UK, *n* = 30); for example, social workers and educators in the UK recommended an opt-out option for parental consent.

There were divergent perspectives on who would have access to the information entered into a risk calculator. Most adolescents in Brazil and Nepal stated that the calculator would only be completed honestly if adolescents knew the information would be kept confidential from their peers, teachers, university admission programs and future employers (BR, NP, *n* = 9). Without proper confidentiality, they were concerned the risk calculator could be stigmatizing for those labeled as high risk (NG, NP, UK, *n* = 48).

As an alternative to adolescents completing the risk calculator, teachers in Nigeria said they would prefer to complete the risk calculator themselves with the information they knew about their students (NG, *n* = 30). Another option was that parents would complete the risk calculator about their children (NP, *n* = 7). However, most respondents did not support parental completion of the risk calculator because they felt the questions on parental relationships and exposure to abuse would not be answered honestly (NG, NP, UK, *n* = 35). Social workers and health-care practitioners in Nepal and Nigeria reported that if issues such as maltreatment were endorsed, this needs to be reported to protective agencies (NG, NP, *n* = 23).

For disclosure of results, there were mixed perspectives. Most participants wanted adolescents to get the results themselves and then adolescents could make behavior changes or do something to mitigate their risk. However, across all countries there were adults who recommended against youth having access to the results because this could lead to negative responses including further increasing the risk of depression. Being told one is high risk could result in "a self-fulling prophecy" triggering depression (BR, NG, NP, UK, *n* = 34). Some adults wanted adolescents to access the recommended care but not disclose the personal risk level to them (NG, NP, *n* = 27). Adolescents in Nepal suggested that they should not have access to the results, and only their parents and teachers would receive the risk scores (NP, *n* = 7). These adolescents felt that this would influence caregivers to change their behavior positively. Respondents in Nigeria said parents would behave better toward the adolescents if they knew risk status (NG, *n* = 19). Some respondents in Nepal stated that parental access to the information could lead parents to view high-risk children more negatively because of the future detrimental consequences of depression (NP, *n* = 3).

There were varied opinions about sharing results with teachers and other stakeholders. Some Nigerian participants reported that teachers could use this to improve their treatment of at-risk adolescents (NG, *n* = 19). Conversely, one Nigerian teacher said it would negatively impact the teachers' treatment of and investment in their students (NG, *n* = 1). In Nepal and Nigeria, there was discussion of policymakers and officials having access to aggregate information to identify schools or regions with high prevalence of at-risk youth (NG, NP, *n* = 28). Private health insurance companies were mentioned as one group that should absolutely not have access to risk information (NP, NG, UK, *n* = 44).

### Actionable

For prediction models to be used, the results had to be actionable. Respondents reported that adolescents, families, schools, health-care practitioners, policymakers and others need to be able to do something about the prediction information (BR, NP, UK, *n* = 96). If an adolescent or parent were told of high-risk status but not provided with prevention resources, this would be potentially more

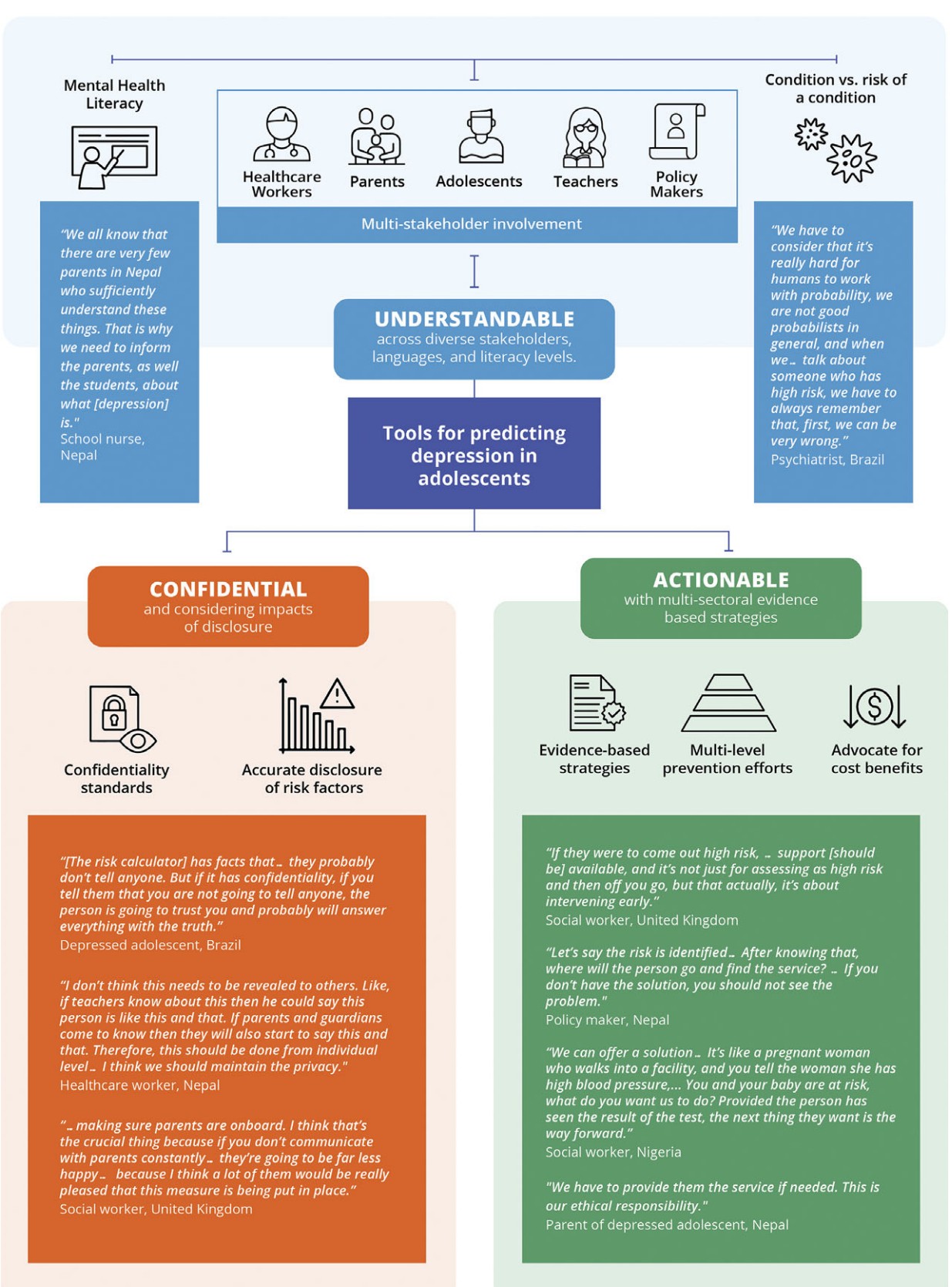

**Figure 1.** Key themes for requirements of implementing risk prediction tools for depression among adolescents based on qualitative findings from Brazil, Nepal, Nigeria and the United Kingdom (*n* = 241 participants).

harmful than not knowing the risk (BR, NP, UK, $n = 43$). Respondents pointed out that prevention strategies needed to be evidence-based for their particular setting (BR, NP, $n = 11$). Nepali respondents expressed a need for multi-level information on prevention, such as (a) what can the individual do on their own, (b) what can the family do, (c) what can educational and health-care institutions do and (d) at the systems level, what can policymakers do (NP, $n = 18$).

Many participants reported they would not know what to do if they received information that they, or their child, were at risk. There were a few instances, specifically in the UK, where respondents felt they would know how to act upon the risk information provided. Responses from adolescents and other stakeholders in Nepal revealed an assumption that if an adolescent had high risk, they could simply go to a counselor or mental health professional who would tell them how to prevent depression (NP, $n = 11$). However, in all countries, health-care practitioners and policymakers pointed out that there were no evidence-based prevention services in their settings (BR, NG, NP, UK, $n = 46$). There was concern that adolescents would look online at YouTube, TikTok and other social media, where they would be directed to prevention strategies that are not evidence-based (BR, NG, NP, UK, $n = 45$). In Nigeria, respondents felt that religious leaders would be able to support those at risk (NG, $n = 3$). In the UK, pastoral care could be provided to those at high risk (UK, $n = 1$).

At a national level, respondents stated that prevention resources needed to be available throughout the country (NP, NG, UK, $n = 35$).

Many respondents worried that even if evidence-based prevention strategies existed, there would not be resources to implement the strategies. Health-care workers and other stakeholders in all countries mentioned that resources for prevention should not detract from resources for treatment (BR, NG, NP, UK, $n = 62$). Respondents brought up that the resources for treatment were already scarce. In the UK, there was concern about austerity measures and cutbacks in mental health care, which would impede prevention services (UK, $n = 43$). They also said that the British education system could not take on prevention services because teachers were overburdened. In Brazil, respondents shared that the government child and adolescent psychiatry services were already unable to meet treatment demands, leaving them unable to do anything for prevention (BR, $n = 12$). In Nepal, adults were concerned adolescents would not utilize preventive services even if they were available (NP, $n = 8$).

## Discussion

To understand attitudes toward use of prediction models to estimate risk of developing depression during adolescence, we conducted a qualitative study with adolescents and other stakeholders in Brazil, Nepal, Nigeria and the UK. Participants identified potential benefits, such as being able to take action to prevent onset of depression. Some health-care providers felt that a depression prediction tool could help them be more objective when identifying adolescents needing services. However, there were a number of

---

**Box 2:** Recommendations on research, practice, and policy for adolescent depression risk prediction tools.

1. ***Ensure patient and public involvement in prediction research***: Prediction research should involve patients and the public, in particular advisory boards comprising adolescents with lived experience of depression. These boards could help interpret and publicly communicate results of prediction research. Moreover, adolescents and experts by experience could advise on what types of risk factors they would feel comfortable disclosing to determine which domains could be included when developing risk prediction tools. Adolescent advisors could also provide perspectives on the wording of risk questions to identify terminology less likely to contribute to inconsistent responses and therefore optimize reliability.

2. ***Identify usage settings and evaluate strategies for obtaining and disclosing risk status for adolescent depression:*** The feasibility and acceptability of using risk prediction tools will depend upon the setting in which they are used. Stakeholder input is necessary to select the proposed setting, even at the stage of developing the risk prediction tool because this will influence the domains included. Different utilization approaches, such as pre- and post-test counseling for HIV or genetic testing, could be adapted and evaluated for depression risk assessment and disclosure. The curriculum needed for someone to take on depression risk counseling could be developed and piloted in different global settings to explore acceptability, feasibility and safety when working with youth to assess and disclose risk status.

3. ***Enhance depression literacy among adolescents and other stakeholders:*** Reducing misconceptions about mental illness and increasing understanding of how it can be treated and managed was considered vital not only for youth but also for teachers, parents and health-care professionals. This included activities such as involving youth who have experience of depression to educate the public. Risk prediction efforts need to integrate a literacy component to facilitate informed uptake of a risk prediction tool. Health literacy related to understanding concepts of risk and probability are also essential for clinicians and policymakers to effectively integrate risk and communicate information to adolescents and caregivers.

4. ***Develop evidence-based and affordable prevention programs for low- and middle-income countries and other low-resource settings***: Efforts to improve prediction should occur hand-in-hand with developing prevention measures that are feasible in low-resource settings. Given that the vast majority of adolescents live in low- and middle-income countries, different options and multilevel approaches for prevention should be tested in these settings including efforts targeting what adolescents could do, what families could do, what schools could do and what societal-level programs policymakers could implement. The evidence could be informed not only by studies demonstrating long-term reduction in incidence of depression but also by trials demonstrating reduction in risk levels. Adolescents', caregivers', health-care providers' and other stakeholders' willingness to use and honestly respond to risk assessment tools is tied with the perceived benefit of what that disclosure will yield. If there is not a perceived benefit in terms of service provision or other prevention guidance, this could severely constrain usage of a tool.

5. ***Develop prevention strategies that do not require disclosure of individual risk profiles:*** Risk scores could be aggregated to help policymakers allocate prevention resources at a school, neighborhood, community or regional level, instead of having to disclose high- versus low-risk results to individual adolescents. Communities prioritized for prevention efforts could be identified based on clustering of key risk factors, for example, school dropout, disciplinary measures related to bullying and fights, child maltreatment and harmful substance use. This would be less stigmatizing than the individual level reporting and also potentially more appropriate given variability in accuracy of prediction models. National and regional surveys that collect related information, for example, the UNICEF Multiple Indicator Cluster Survey, could be used in national programs to target the settings where prevention programs most need to be established.

6. ***Assure investment in prevention that is additional to investment in treatment:*** Prevention efforts are unlikely to be implemented if they are perceived as diverting resources from existing treatment programs. Therefore, funding prevention requires additional allocation. The future economic benefits of prevention efforts require cost analyses to encourage action by policymakers. Given that multivariate risk scores also provide continuous outcomes, policymakers could rationalize limited prevention resources at higher risk thresholds initially, then lower the thresholds for prevention as more resources or more affordable prevention strategies become available. Additionally, with the long period of time needed to demonstrate benefits of investment in prevention programs, changes in risk scores could be a shorter term proxy for estimating cost savings in the future. As prediction research grows, research funders should ensure prevention efforts are financed and advanced at a comparable pace.

concerns. First, stakeholders reported low levels of mental health literacy in their communities for understanding depression versus risk of depression. Second, there was concern that educators, employers and health insurance companies could access risk information. Some parents and teachers felt that adolescents themselves should not know their risk because of creating a "self-fulfilling prophecy" that could cause depression. The third concern was the need for risk information to be actionable: if someone is told about their risk, then they need information on how to reduce the risk or other actions they could take. However, many stakeholders were not aware of prevention services in their settings. Teachers, clinicians and policymakers said resources for prevention were insufficient and financing prevention could jeopardize the already limited funding for treatment services.

Some findings differed from other research on attitudes toward prediction. In prior studies, participants often reported a desire to know their risk results (Bellón et al., 2014, Bui et al., 2014, Welsh and Tiffin, 2011). In the PredictD qualitative study in Spain, adults in primary care welcomed the knowledge about their depression risk and felt it gave them an opportunity to talk with their primary care doctor about prevention (Bellón et al., 2014). In other studies with results supporting disclosure of risk status, the participants already had a known risk factor, most commonly a family history of severe mental illness, or they had subsyndromal symptoms, such as for psychosis (Welsh and Tiffin, 2011). Others were adult volunteers in biological research, such as genetic studies (Bui et al., 2014).

However, other studies have reported participants' hesitation about receiving risk information. Among patients in German early detection centers, one third did not want predictive information when offered a hypothetical opportunity to learn about future risk (Mantell et al., 2021). One methodological finding in prior research is that the more realistic prediction scenarios are, the less likely participants are to want to receive risk results (Alder et al., 2013). Given that the risk calculator in our study was presented as a tool based on research conducted in the participants' settings, completing the calculator may have felt like a real-world scenario and thus increased hesitation to know one's risk status. Ultimately, exploring attitudes versus actual behaviors related to using risk prediction would need to be tested in longitudinal trial conditions. For example, in the PredictD study in Spain, primary care patients wanted to know their depression risk, but physicians expected patients would not want this information (Moreno-Peral et al., 2019). Then, when tested in the context of delivering the prediction and an associated preventive intervention, clinicians saw that assessing risk and administering prevention efforts led to decreased anxiety on the part of patients over 18 months (Moreno-Peral et al., 2021).

Participants in our study reported lack of mental health literacy as a barrier to implementation of prediction models. Lower mental health literacy has been associated with reluctance to know one's risk (Mantell et al., 2021). Moreover, stigma impedes prevention programs, with self-stigmatization and self-labeling observed among persons identified as at-risk for psychosis (Rüsch et al., 2015, Rüsch and Thornicroft, 2014). Other studies supported our finding of the perception that knowing one's risk could trigger a "self-fulfilling prophecy" of developing a psychiatric condition (Mantell et al., 2021). Taken together, these points highlight the need for careful consideration of the context and setting for implementing risk prediction tools in order to minimize stigma, anxiety and other negative personal and social ramifications.

An issue raised by our participants was the challenge of understanding risk probability. Other researchers have raised concerns about the public's interpretation of prediction probabilities (Lawrie et al., 2019), and physicians also poorly interpret probability in clinical practice (Whiting et al., 2015). It is important that future studies assess how risk prediction models of depression are communicated to lay and clinical audiences in ways that are easily understood. This begins with communicating concepts such as relative risk, absolute risk, risk difference, calibration and discrimination to clinicians and policymakers, thus enabling them to communicate clearly to adolescents and their caregivers to make informed treatment decisions. Some strategies for adolescents and caregivers could be adapted from visual analogues used in genetic counseling (Austin, 2020). A number of online tools to illustrate probability have been developed, such as risk for heart attack or stroke using QRISK®3 (www.qrisk.org) (Hippisley-Cox et al., 2017). In addition, clinicians and policymakers need to understand the samples on which risk prediction models are calibrated to judge applicability to a particular adolescent. The good news from our discussions with policymakers was the interest to know how accurate these models were for their communities before deciding to implement the tool. Interactive visualizations developed by the DEPRESSD Project (www.depressd.ca/tools) to help clinicians understand how depression screening tools perform in their setting could be adapted for contextualizing interpretation of depression risk results (Levis et al., 2019).

The overarching finding of this qualitative work was risk prediction needed to be done in contexts where knowledge was actionable through clinicians, caregivers, adolescents and others having evidence-based strategies to mitigate the risk. Actionability has been raised previously as a major ethical concern in prediction models (Lawrie et al., 2019). Currently, depression prevention interventions have small effect sizes, and the majority of prevention trials are universal (given to everyone) or indicated (provided to individuals already displaying symptoms) (Conejo-Cerón et al., 2017, Hetrick et al., 2016, Mendelson and Eaton, 2018). Given that prediction models are based on risk factors, there is also a need for evidence on selective interventions, which target persons with known risk factors. In a review of school-based programs for adolescent depression prevention, only 5 of 40 prevention programs were selective (Werner-Seidler et al., 2017). Among these, the risk factors varied widely, e.g., personality, living in a low-income area, externalizing symptoms and exposure to political violence. In a review of cognitive behavioral therapy used for prevention of depression among adolescents, only 2 out of 23 studies were purely selective prevention (Rasing et al., 2017). In a more recent review, 6 out of 14 prevention trials for adolescents were selective, with most using parental history of depression as the sole risk factor, but only 1 trial showed benefits (Cuijpers et al., 2021). A 2016 Cochrane review of prevention of adolescent depression using psychological interventions reported "a sobering lack of effect" for universal prevention and concluded "that there is still not enough evidence to support the implementation of depression prevention programs" (Hetrick et al., 2016). This highlights the need for more efforts in prevention research.

Our participants were concerned about the costs of prevention services. A review of cost-effectiveness of prevention did not find data on any trials that were exclusively selective interventions for adolescents (Conejo-Cerón et al., 2021). Moreover, when evidence-based selective prevention interventions are identified, the length of time needed to show the benefit of prevention may be a disincentive for investment by term-limited politicians and policymakers (Duevel et al., 2020, Knapp and Wong, 2020). Taken together, this raises

concern that the field of prediction may outpace the development, testing and funding of selective prevention interventions.

Given the findings in this study, we recommend a path for adding qualitative components to prediction research. Box 2 lists six recommendations for potential applications of our study methods and findings for prediction research. The ethical concerns and societal implications are even more relevant when considering the exponential increase in prediction modeling through use of artificial intelligence with big data using medical records, online personal information and mobile technology tracking personal behavior (Balliu et al., 2024, Koutsouleris et al., 2018, van Dellen, 2024).

### Strengths and limitations

This is the first study, to our knowledge, to identify attitudes toward risk detection of depression with a focus on youth in LMICs. Moreover, this is the first study to use a risk calculator prototype that had been validated in the study settings, thus representing information that could be implemented in their settings. Our study has a number of limitations. Notably, the samples in Nigeria and the UK did not include direct participation of adolescents. Further research is needed to obtain more comprehensive perspectives directly from adolescents. The attitudes collected here are also limited to demographic and social risk factors. Attitudes may differ if biological markers, digital phenotypes, or other types of risk information were used. Prediction models using online information or passive sensing from personal digital devices would likely accentuate confidentiality concerns.

### Conclusion

Although accurate risk prediction is an objective to reduce the global burden of depression, the potential benefits of risk classification can only be realized if adolescents and associated stakeholders are willing to participate in using risk calculators and similar tools. This engagement is contingent upon increasing public health literacy of depression, understanding what risk prediction means in lay terms, being certain that information will be confidential, and, most importantly, knowing that adolescents identified as high risk will be supported with evidence-based and affordable prevention resources.

**Open peer review.** To view the open peer review materials for this article, please visit http://doi.org/10.1017/gmh.2024.136.

**Supplementary material.** The supplementary material for this article can be found at http://doi.org/10.1017/gmh.2024.136.

**Data availability statement.** Anonymized code queries are provided in Supplemental File 3.

**Acknowledgments.** We are grateful to the individuals who participated in this study and to all members of the IDEA team for their dedication, hard work and insights. Figure is designed by Cheenar Shah.

**Author contribution.** CK and HLF are the joint senior authors; Conceptualization: BAK, VM, AA, CK, HLF; Methodology: BAK, SSW, VM, AA, CK, HLF; Formal analyses: BAK, SSW, KO, AB, AV, TM, SB, OM, JB, VN; Investigation: KO, AB, AV, TM, SB, OM, JB, VN; Resources: CK; Data curation: SSW, AB, AV, OM, JB, VN; Writing – original draft: BAK, SSW, KO; Writing – review and editing: AB, AV, TM, SB, OM, JB, VN, KG, AA, VM, CK, HLF; Visualization: BAK; Supervision: BAK, SSW, KG, AA, VM, CK, HLF; Project administration: BAK, VM, CK, HLF; Funding acquisition: VM, CK.

**Financial support.** The IDEA project is funded by an MQ Brighter Futures grant [MQBF/1 IDEA]. Additional support was provided by the UK Medical Research Council [MC_PC_MR/R019460/1] and the Academy of Medical Sciences [GCRFNG\100281] under the Global Challenges Research Fund. Prof. Fisher receives support from the Economic and Social Research Council (ESRC) Centre for Society and Mental Health at King's College London [ES/S012567/1]. Dr. Kieling is a Conselho Nacional de Desenvolvimento Científico e Tecnológico (CNPq) researcher and an Academy of Medical Sciences Newton Advanced Fellow. Prof. Valeria Mondelli is supported by the National Institute for Health and Care Research (NIHR) Maudsley Biomedical Research Centre at South London and Maudsley NHS Foundation Trust and King's College London. The views expressed are those of the authors and not necessarily those of the NHS, the NIHR, the Department of Health and Social Care, the ESRC or King's College London.

**Competing interest.** Prof. Mondelli has received research funding from Johnson & Johnson, a pharmaceutical company interested in the development of anti-inflammatory strategies for depression, but the research described in this paper is unrelated to this funding. All other authors declare they have no conflicts of interest to report.

**Ethics statement.** The study was approved by the Institutional Review Board of George Washington University, USA, the Nepal Health Research Council in Nepal, the Ethics Committee at Hospital de Clínicas de Porto Alegre in Brazil, the Lagos State University Teaching Hospital Research and Ethics Committee and the Research and Ethics Committee of the Federal Neuropsychiatry Hospital Yaba, Lagos, in Nigeria, and the Psychiatry, Nursing and Midwifery Research Ethics Subcommittee at King's College London in the UK. All participants were provided with information on mental health services and psychosocial support programs, and any participants reporting distress during the interview process were referred to a study counselor or other mental health professional for immediate support.

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
