## [Reviewer Report]

This is an interesting paper addressing an important issue – the views of key stakeholders about a youth depression risk prediction tool. Given the interest in risk prediction in psychiatry and the challenge in ensuring risk prediction tools are feasible and acceptable and possible to implement in practice, this is important. Overall, the paper is clearly written, and the results are clear. However, I have several suggestions about how the manuscript can be improved. In particular, the arguments presented about risk prediction as relate to depression in different contexts require some nuance and revision. Moreover, the recommendations presented in panel 2 seem to be somewhat overstated. I provide detailed suggestions below:

1. Abstract. Review and consider the use of the word disease to describe depression. Some readers may find this term stigmatizing.

2. Introduction. It would be helpful for the reader to explain the purpose of risk prediction early in the introduction i.e. using information about an individual’s future risk of a condition to guide clinical decision making and inform intervention choice. (I.e. risk prediction is really only useful if it can be used in a clinical setting) which somewhat mirrors the findings of this qualitative study.

3. Introduction & impact statement. The increased uptake of machine learning, AI and big data is used as an introductory rationale for an interest in risk prediction. However, the risk prediction calculator presented in the paper is based on simple, easy to assess predictors. Moreover, none of the cited references specifically refer to using “big data” techniques to predict depression. Please consider removing that part of the introduction and presenting it as a discussion point instead. It seems at odds with the rationale for the current study. As an alternative introductory point, the authors could potentially mention the routine use of risk prediction calculator in clinical practice in other areas of medicine e.g. cardiovascular disease and highlight the potentially more complex ethical issues around acceptability, feasibility and implementation as a rationale for the current study. Please consider.

4. Introduction. It is mentioned that many risk prediction tools in psychiatry focus on specific high-risk groups e.g. those with a family history of a condition. It is worth explaining why this is done. Also, there is a published risk prediction calculator of adolescent depression in this population which has been developed and validated in two UK samples. Please cite that paper: Stephens A, Allardyce J, Weavers B, Lennon J, Jones RB, Powell V, Eyre O, Potter R, Price VE, Osborn D, Thapar A, Collishaw S, Thapar A, Heron J, Rice F. Developing and validating a prediction model of adolescent major depressive disorder in the offspring of depressed parents. J Child Psychol Psychiatry. 2023 Mar;64(3):367-375. doi: 10.1111/jcpp.13704. Epub 2022 Sep 12. PMID: 36096685; PMCID: PMC10087673.

It may be worth considering citing the review by Austin on risk and interpretation of risk in psychiatric research both in the introduction and the discussion. Austin JC. Evidence-Based Genetic Counseling for Psychiatric Disorders: A Road Map. Cold Spring Harb Perspect Med. 2020 Jun 1;10(6):a036608. doi: 10.1101/cshperspect.a036608. PMID: 31501264; PMCID: PMC7263094.

5. In presenting the risk calculator that the present study is based on it is stated “although these findings are under the recommended cut-off of 0.80 AUC…..it is foreseeable that further refinement will result in improved accuracy in the future….” While this may be true, it is also the case that there are well documented reasons for non-transportability of risk prediction findings across contexts (e.g. Altman). This will include differences in the measurement and meaning of predictors in different settings, different case mixes and, for depression, potentially measurement differences including cultural ones in how depression is assessed and understood. This is important to note not only for its own sake but also because it highlights the value of undertaking cross cultural studies (as done in this paper) and highlights the importance of different settings for the validity of the model but also future implementation where clinical practices, resources and knowledge may differ. Finally, please mention the time period over which risk for depression is predicted. I could not see this but it is relevant for potential implementation of a risk calculator.

6. Methods – only in this section is the context of healthcare services (and differences across countries) mentioned. Please mention this in the introduction briefly (this also relates to point 5).

7. Methods – it would be helpful if the authors could briefly mention the sites where the data collection took place e.g. rural:urban etc to give the reader some idea about how the site relates to the description of the country.

8. Methods – please provide more detail on the policy maker category. It would be helpful to have a description of who these people are and at what level they make policy decisions or in what context. E.g. local, education?

9. Discussion. I would suggest including some additional general information to help the reader appropriately interpret the meaning of the results presented here in the broader context of risk prediction. In particular, please mention in the discussion the following points: A) the need for a careful consideration of context and setting for implementation. This is an important point to include as it is so important for successful implementation (that is helpful rather than stigmatizing or anxiety provoking). B) the relationship between attitudes about knowing personal risk and whether that knowledge is “actionable” i.e. service availability. For instance, it could be argued that there is little value to an individual in knowing details about individual risk unless it can be a route to support of some kind. C) Some technical aspects about risk prediction are relevant to usefulness and understanding. These include the metric used to provide risk information i.e. absolute rather than relative risks are preferred (see e.g. Austin; Uher) and the accuracy of risk prediction (i.e. the calibration of a risk calculator (as opposed to just the ability to discriminate which is what AUC is based on). One point the authors may consider including relates to the point about people being unsure about whether they would like to know their individual risk information: even without this information, people often have heuristics about their level of risk that may or may not be reliable (e.g. people with a family history of a condition often overestimate their absolute risk) – Austin discusses this point clearly in her review. Finally, it seems worth citing the Cochrane review of CBT preventive interventions (Heterick et al. 2016) but crucially also considering the wider value of risk prediction for prevention e.g. different approaches may be valuable (see suggestions in section 9 (on recommendations 4-6)). Please consider incorporating some of this information into the discussion.

10. The research recommendations in panel 2 seem somewhat overstated and do not appear to be fully justified by the findings of this study. Many of the recommendations are mentioned in the risk prediction literature and in the prevention literature. Moreover, given the purpose of risk prediction to aid clinical decision making, the fact that no clear clinical setting has been identified for this tool may preclude clear recommendations being made. I would argue that considering the clinical setting where a tool could be implemented is an important aspect of risk prediction that needs to be considered with key stakeholders at an early stage. The recommendations in their current form are also long and very detailed. Please re-consider some of these and write in more general terms ensuring that the specific details of this risk prediction calculator are reflected on and used to make the recommendations. Specific points relating to the recommendations:

Recommendation 1 “ensuring PPI in prediction research” – this is a crucial aspect of ensuring acceptability, feasibility and successful implementation of any risk prediction model. This is understood in the wider risk prediction literature. Please write this in a broader way to reflect this.

Recommendation 2: “understanding depression” Only enhancing depression understanding is recommended. However, much work needs to be done also about understanding and explaining risk as found in this study. See e.g. Austin in the review about psychiatric genetic counselling and explaining risk.

Recommendation 3: “ evaluate different strategies for obtaining and disclosing risk status…” This seems somewhat premature. Use of the term evaluate implies comparison. I would argue what is required first is a consideration of the clinical (e.g. health/education etc) settings in which a tool could be evaluated. Moreover, this recommendation clearly relates to the type of information that was included in the risk calculator as predictor variables (many of which were personal and highly confidential). An alternative approach, would be to select different predictor variables where these concerns are lessened. Please reconsider this in a broader way, reflecting on the specifics of the tool presented here.

Recommendations 4-6 relate to prevention. Many of these points relate to policy/political decisions about the approach to prevention that is taken. For instance, one policy decision based on the results of the risk prediction calculator presented here (and other information/research) would be to select important risk factors and aim to reduce these at a population level (e.g. child maltreatment). A different approach would be to “rationalize” prevention resources and allocate them to those at high-risk. Yet another, would be to ensure that those with current depression or imminent risk of depression get timely access to evidence-based treatment. Please think these recommendations over and specify them in more general terms, reflecting on the specific details of this risk prediction calculator.

---

## [Reviewer Report]

Thanks for letting me review this paper whose main objective aims to explore the attitudes of different stakeholders, across diverse cultural settings and in countries that meet the entire range of income levels, towards the use of predictive models for identifying the risk of depression during adolescence. The text is clear and well written. In general the qualitative data have been analysed appropriately, the results clearly described, the figures and tables suitable, clear and consistent with the text. The conclusions are supported by the evidence and their conclusions would help different stakeholders (adolescents, parents, teachers, healthcare practitioners, and policymakers) make decisions on the application and use of risk algorithms to predict the onset of depression episodes in adolescents. With the aim of improving the paper, I will point out some suggestions and questions:

One of the strengths of this study is the possibility of checking contradictions between the perspectives and opinions of different stakeholders, taking into account the different cultural contexts of the respective countries. Maybe the authors could go into more depth in discussing these aspects. For example, when a group of stakeholders expresses opinions on possible side effects of using information on the risk of depression in adolescents (fear, anxiety, worsening of emotional well-being, etc.), these are speculations that would have to be verified from the point of view of scientific evidence through clinical trials, comparing the intervention group (in which predictive algorithms are used) and the usual care control group without the use of algorithms. This also occurred in a qualitative study (1) on whether adult patients wanted to know their risk of depression in the future and whether GPs were willing to communicate this. Patients were delighted to know this and GPs thought that patients would not want to know because it could cause fear and anxiety. It was later found that communicating this risk of depression by GPs and discussing its prevention with their patients reduced the incidence of anxiety disorders during the 18 months of the trial follow-up (2).

(1) Moreno-Peral P et al. Family physicians' views on participating in prevention of major depression. The predictD-EVAL qualitative study. PLoS One. 2019 May 30;14(5):e0217621.

(2) Moreno-Peral P, et al. Use of a personalised depression intervention in primary care to prevent anxiety: a secondary study of a cluster randomised trial. Br J Gen Pract. 2021 Jan 28;71(703):e95-e104

Other aspects that could be discussed in the manuscript are the following:

Some questions on the depression risk calculator may be “hot questions” that are difficult to answer because of their high emotional content and family and social implications. For example, “did someone ever try to touch you in sexual way or ask you to touch them against your will?” This kind of questions could potentially be a good predictor of depressive episodes; however, their answers may be unreliable. Before deciding to expose the adolescent population to these questions to test predictive validity, it would be necessary to check whether their answers are reliable (e.g. by doing a test-retest reliability analysis on stability over time, asking them again 7-10 days later to a sample of adolescents). Possibly, this type of questions could also be related to the resistance and doubts raised by some groups of interviewees.

Interview participants were able to click through the risk calculator entering dummy information; however, no score was calculated for the participants. As the authors say, this would be useful to avoid adverse effects on participants. However, especially in adolescents and parents, it could condition their opinions on what they fear could happen rather than on what actually happens.

It is true that many patients find it difficult to understand the probability of risk. And many clinicians also find it difficult to explain it to their patients. This is undoubtedly a major barrier to the widespread use of risk algorithms in practice. The way in which the risk is communicated can also influence its preventive success. For example, excessive emphasis or the use of words with a high emotional content could have the opposite effect to that desired. There are some ways of communicating risk that may be more efficient, for example using intuitive graphics (see e.g. how QRISK3 does it for cardiovascular risk: https://qrisk.org/). It is also important that at the same time as informing-knowing the risk, information is available on things that the adolescent could do to reduce that risk, which are feasible and realistic in their context (cultural and health services) and which are supported by scientific evidence.

---

## [Reviewer Report]

The authors have responded appropriately to all my suggestions and recommendations. I sincerely believe that this manuscript provides important contributions to knowledge in the field of applying risk algorithms to predict the onset of depressive episodes and initiate their prevention in adolescents.

---

## [Editor Report]

Dear Authors,

Your revised manuscript titled “No prediction without prevention: A global qualitative study of attitudes towards using a prediction tool for risk of developing depression during adolescence” has been reviewed